# Dynamical Localization Simulated on Actual Quantum Hardware

**DOI:** 10.3390/e23060654

**Published:** 2021-05-23

**Authors:** Andrea Pizzamiglio, Su Yeon Chang, Maria Bondani, Simone Montangero, Dario Gerace, Giuliano Benenti

**Affiliations:** 1Dipartimento di Scienza e Alta Tecnologia, Università degli Studi dell’Insubria, Via Valleggio 11, 22100 Como, Italy; apizzamiglio@studenti.uninsubria.it; 2Institute of Physics, Ecole Polytechnique Fédérale de Lausanne (EPFL), 1015 Lausanne, Switzerland; su.chang@epfl.ch; 3CERN, 1211 Meyrin, Switzerland; 4Istituto di Fotonica e Nanotecnologie, Consiglio Nazionale delle Ricerche, Via Valleggio 11, 22100 Como, Italy; maria.bondani@uninsubria.it; 5Dipartimento di Fisica e Astronomia “G. Galilei”, Università di Padova, 35131 Padova, Italy; simone.montangero@unipd.it; 6Istituto Nazionale di Fisica Nucleare, Sezione di Padova, 35131 Padova, Italy; 7Padua Quantum Technology Research Center, Università di Padova, 35131 Padova, Italy; 8Dipartimento di Fisica, Università di Pavia, Via Bassi 6, 27100 Pavia, Italy; dario.gerace@unipv.it; 9Istituto Nazionale di Fisica Nucleare, Sezione di Milano, Via Celoria 16, 20133 Milano, Italy; 10NEST, Istituto Nanoscienze-CNR, 56126 Pisa, Italy

**Keywords:** digital quantum simulation, quantum algorithms, complex quantum systems

## Abstract

Quantum computers are invaluable tools to explore the properties of complex quantum systems. We show that dynamical localization of the quantum sawtooth map, a highly sensitive quantum coherent phenomenon, can be simulated on actual, small-scale quantum processors. Our results demonstrate that quantum computing of dynamical localization may become a convenient tool for evaluating advances in quantum hardware performances.

## 1. Introduction

The simulation of complex systems is expected to be one of the main applications of future quantum computers [1]. The special role of quantum mechanics in simulation was pointed out long time ago by Feynman [2]. He observed that, while in a classical computer the memory requirements for the simulation of a many-body quantum system can grow exponentially with the number of particles, the growth is only polynomial in a quantum computer, which is itself a many-body quantum system. Lloyd [3] then validated the conjecture of an exponential speedup in the simulation of quantum systems by means of a quantum computer, with respect to a classical computer. Applications have been discussed for many physical models, so far (see, e.g., Refs. [4,5,6,7,8,9,10,11,12,13], and references therein). An ideal quantum computer operating with more than fifty qubits could outperform a classical computer, and the quantum advantage for specific problems has been recently claimed [14,15]. However, quantum advantage can only be reached with a high enough quantum gate precision and through processes generating a large enough amount of entanglement, otherwise they can be efficiently simulated via tensor network methods [16]. On the other hand, present-day quantum computers suffer from significant decoherence and the effects of various noise sources, such that the amount of entanglement that can be reached is still limited. Therefore, achieving the quantum advantage in complex, practically relevant problems such as chemical reactions, new materials design, or biological processes, is still an imposing challenge. It is then important to benchmark the progress of currently available quantum computers by simulating less demanding but still physically significant tasks.

Dynamical localization is one of the most interesting phenomena that characterize the quantum behavior of classically chaotic systems: quantum interference effects suppress the diffusion taking place in the underlying classical model, leading to exponentially localized wave functions. This phenomenon, first discovered in the quantum kicked-rotor model [17,18], has been observed experimentally in the microwave ionization of Rydberg atoms [19], in atom-optic systems [20,21,22], and in nuclear magnetic resonance quantum ensemble computation [23]. Dynamical localization has deep analogies with Anderson localization of electronic transport in disordered materials [24]. Localization is indeed ubiquitous in wave physics, since it originates from the interference between multiple scattering paths. It has been shown that in a quantum computer simulating dynamical localization the amount of entanglement is deeply connected with the localization length of the system being simulated [25].

In this paper, we simulate dynamical localization with three qubits by means of several IBM quantum processors, with the aim of analyzing the performances of current noisy quantum hardware, freely available for cloud quantum computing. In particular, we use the known quantum algorithm for the quantum sawtooth map [26,27], which is the ideal model to investigate dynamical localization, and more generally quantum chaotic dynamics on a quantum computer. This is because all of the qubits are actually used to simulate dynamics, without the need for auxiliary qubits. Moreover, the quantum algorithm performs forward-backward Fourier transform, thus exploring the entire Hilbert space of the quantum register in a complex multiple-path interferometer that leads to wave-function localization. As such, dynamical localization is a very fragile quantum phenomenon, extremely sensitive to noise. Its effective simulation on an actual quantum hardware is therefore a potentially ideal test bench to illustrate the power of quantum computation.

Here we show that it is possible to detect quantum localization with n=3 qubits and a single step of the sawtooth map, requiring 4n=12 single-qubit and $2(n^{2}-n)=12$ two-qubit quantum gates. We find that the height of the localization peak is smaller than the value expected for a noiseless quantum computing machine. Moreover, the peak reduction in real quantum processors is significantly larger than expected from the Qiskit simulator, which only takes into account a limited number of noise channels. Our results show that for the best performing and freely available IBM quantum processors the localization peak does emerge from noise for about 3 map steps.

The paper is organized as follows. In Section 2 we recall the main steps of the quantum algorithm for simulating dynamical localization (see Refs. [26,27] for a detailed description). In Section 3 results obtained from IBM quantum processors are shown and compared with those from the IBM simulator and with the exact, noiseless evolution. Our conclusions are finally drawn in Section 4.

## 2. Quantum Algorithm for the Dynamical Localization

### 2.1. Quantum Algorithm for the Sawtooth Map

The most convenient model to simulate dynamical localization on a quantum computer is the quantum sawtooth map. This map describes the dynamics of a periodically driven system, as derived from the Hamiltonian
(1)$$H\left(\theta ,I;\tau \right)=\frac{I^{2}}{2}+V\left(\theta \right)\sum_{j=-\infty }^{+\infty }\delta \left(\tau -jT\right)\;,$$
where (I,θ) are conjugate action-angle variables (0≤θ<2π), with the usual quantization rules, θ→θ and I→I=−i∂/∂θ (we set ℏ=1) and $V\left(\theta \right)=-k{\left(\theta -\pi \right)}^{2}/2$. This Hamiltonian is the sum of two terms, $H\left(\theta ,I;\tau \right)=H_{0}\left(I\right)+H_{k}\left(\theta ;\tau \right)$, where $H_{0}\left(I\right)=I^{2}\;/2$ is the Hamiltonian of a particle freely moving on a circle parameterized by the coordinate θ, while $H_{k}\left(\theta ;\tau \right)=V\left(\theta \right)\sum_{j}\delta \left(\tau -jT\right)$ represents a force acting on the particle and switched on and off instantaneously (kicking potential) at time intervals *T*. The evolution from time $tT^{-}$ (prior to the *t*-th kick) to time $\left(t+1\right)T^{-}$ (prior to the (t+1)-th kick) is described by a unitary operator *U* acting on the wave function ψ:(2)$$\begin{array}{cc} \; & \psi_{t+1}=U\;\psi_{t}=U_{T}U_{k}\;\psi_{t}\;;\\ \; & U_{T}=e^{-iTI^{2}\;/2}\;,\;\;U_{k}=e^{ik{\left(\theta -\pi \right)}^{2}\;/2}\;. \end{array}$$
This map is known as quantum sawtooth map, since the force F(θ)=−dV(θ)/dθ=k(θ−π) has a sawtooth shape, with a discontinuity at θ=0.

The map (Equation 2) can be efficiently simulated on a quantum computer. The quantum algorithm simulating the sawtooth map dynamics is based on the forward/backward quantum Fourier transform between action (momentum) and angle bases. Such an approach is convenient since the operator *U*, introduced in Equation (Equation 2), is the product of two operators, $U_{k}$ and $U_{T}$, diagonal in the θ and *I* representations, respectively. This quantum algorithm requires the following steps for one map iteration:We apply $U_{k}$ to the wave function ψ(θ). In order to decompose the operator $U_{k}$ into one- and two-qubit gates, first of all we write θ in binary notation:
(3)$$\theta =2\pi \sum_{j=1}^{n}\alpha_{j}2^{-j}\;,$$
with $\alpha_{j}\in \left\{0,1\right\}$. As *n* is the number of qubits, the total number of levels in the quantum sawtooth map is $N=2^{n}$. Using expansion (Equation 3), we can decompose $U_{k}$ into a product of $n^{2}$ two-qubit operations, each acting non-trivially only on the qubits $j_{1}$ and $j_{2}$. In the computational basis $\left\{|\alpha_{j_{1}}\alpha_{j_{2}}\rangle \right\}$ each two-qubit gate can be written as $\exp(i2\pi^{2}kD_{j_{1},j_{2}})$, where $D_{j_{1},j_{2}}$ is a diagonal matrix:
(4)$$D_{j_{1},j_{2}}\;=\;\;\left[\;\begin{array}{cccc} \frac{1}{4n^{2}} & 0 & 0 & 0\\ 0 & -\frac{1}{2n}\left(\frac{1}{2^{j_{1}}}-\frac{1}{2n}\right) & 0 & 0\\ 0 & 0 & -\frac{1}{2n}\left(\frac{1}{2^{j_{2}}}-\frac{1}{2n}\right) & 0\\ 0 & 0 & 0 & \left(\frac{1}{2^{j_{1}}}-\frac{1}{2n}\right)\left(\frac{1}{2^{j_{2}}}-\frac{1}{2n}\right) \end{array}\;\right]\;.$$Neglecting a global phase factor of no physical significance, we can decompose $\exp(i2\pi^{2}kD_{j_{1},j_{2}})$ in terms of quantum gates (P and CP) available in the IBM Qiskit interface: -4.6cm0cm
(5)$$e^{i2\pi^{2}kD_{j_{1},j_{2}}}=\left(\begin{array}{cccc} 1 & 0 & 0 & 0\\ 0 & e^{-\frac{i\pi^{2}k}{2^{j_{1}}n}} & 0 & 0\\ 0 & 0 & 1 & 0\\ 0 & 0 & 0 & e^{-\frac{i\pi^{2}k}{2^{j_{1}}n}} \end{array}\right)\left(\begin{array}{cccc} 1 & 0 & 0 & 0\\ 0 & 1 & 0 & 0\\ 0 & 0 & e^{-\frac{i\pi^{2}k}{2^{j_{2}}n}} & 0\\ 0 & 0 & 0 & e^{-\frac{i\pi^{2}k}{2^{j_{2}}n}} \end{array}\right)\left(\begin{array}{cccc} 1 & 0 & 0 & 0\\ 0 & 1 & 0 & 0\\ 0 & 0 & 1 & 0\\ 0 & 0 & 0 & e^{\frac{i\pi^{2}k}{2^{j_{1}+j_{2}-1}}} \end{array}\right).$$The first two factors in the product correspond to phase-shift gates (P gates) applied to qubit $j_{1}$ and $j_{2}$, respectively, while the last one corresponds to a controlled phase-shift gate (CP gates) applied on both qubits, if $j_{1}\neq j_{2}$, and to a P gate if $j_{1}=j_{2}$. Overall we need then $n^{2}-n$ CP e $2n^{2}+n$ P gates to implement $U_{k}$. As $U_{k}$ is the product of diagonal matrices in the θ representation, the order of P and CP gates is irrelevant. Therefore, changing their order allows combining gates applied to the same qubits, thus improving the efficiency of the quantum algorithm. We then reduce the number of quantum gates needed to implement $U_{k}$ to n(n−1)/2 CP and *n* P gates. We note that the above decomposition for $U_{k}$ is possible thanks to the particular form of V(θ) for the sawtooth map. This is the reason for which the quantum sawtooth map is the most convenient model to simulate dynamical localization. For other potentials, like V(θ)=cosθ corresponding to the kicked-rotor model [17,18], auxiliary qubits are introduced to efficiently compute $U_{k}$ [9], thus increasing the overall simulation cost.The change from the θ to the *I* representation is obtained by means of the quantum Fourier transform (QFT), which requires *n* Hadamard (H) gates and $\frac{1}{2}n\left(n-1\right)$ CP gates. The correct order of qubits after the Fourier transform is obtained by simply relabeling qubits, thus avoiding $\left\lfloor n/2\right\rfloor$ physical SWAP gates.In the *I* representation, the operator $U_{T}$ has essentially the same form as the operator $U_{k}$ in the θ representation and can therefore be similarly decomposed.We return to the initial θ representation by application of the inverse QFT.
Overall, this quantum algorithm requires $O(n^{2})$ gates per map iteration. This number is to be compared with the $O(n2^{n})$ operations required by a classical computer to simulate one map iteration using the fast Fourier transform. The quantum simulation of the quantum sawtooth map dynamics is then exponentially faster than any known classical algorithm. Note that the resources required to the quantum computer to simulate the sawtooth map are only logarithmic in the system size *N*. With the above discussed improvements of the quantum circuit, we need $2(n^{2}-n)$ CP, 2n P, and 2n Hadamard (H) gates to simulate one step of the quantum sawtooth map. Which corresponds to 24 quantum gates (12 single-qubit a 12 two-qubit gates) for the case n=3 which we have simulated on the IBM quantum hardware. The angles used in the *P* gates are obtained by changing order of the gates and combining them. The explicit formulas for the angles are given as the following:(6)$$\theta_{k}^{j}=-\frac{2\pi^{2}k}{2^{j}}+\frac{\pi^{2}k}{2^{2j-1}},\;\theta_{T}^{j}=\frac{2N^{2}T}{2^{j+2}}-\frac{N^{2}T}{2^{2j+1}},\;\phi_{k}^{j_{1},j_{2}}=\frac{2\pi^{2}k}{2^{j_{1}+j_{2}-1}},\;\phi_{T}^{j_{1},j_{2}}=-\frac{2N^{2}T}{2^{j_{1}+j_{2}+1}}.$$
The circuit (see Figure 1) avoids SWAP gates via relabeling of qubits after the QFT and the inverse QFT.

### 2.2. Dynamical Localization for the Sawtooth Map

The classical limit of the quantum sawtooth map is obtained for k→∞ and T→0, at constant K=kT. The classical motion is chaotic for K<−4 and K>0. Although the sawtooth map is a deterministic system, in this regime the motion along the momentum direction is in practice indistinguishable from a random walk, with diffusion in the momentum variable. If we consider a classical ensemble of trajectories with fixed initial momentum $m_{0}$ and random initial angle θ, the second moment of the distribution grows linearly with the number *t* of map iterations,
(7)$$\langle {\left(\Delta m\right)}^{2}\rangle \approx Dt,$$
with the diffusion coefficient $D\left(k\right)\approx \left(\pi^{2}/3\right)k^{2}$ for K>1.

The quantum sawtooth map, in agreement with the correspondence principle, initially exhibits diffusive behavior, with the classical diffusion coefficient *D*. However, after a break time $t^{⋆}$, quantum interference leads to suppression of diffusion. For $t>t^{⋆}$ the quantum distribution reaches a steady state which decays exponentially over the momentum eigenbasis:(8)$$W_{m}\equiv |\langle m|\psi \rangle {|}^{2}\approx \frac{1}{\ell }\;\exp\;\left[-\frac{2|m-m_{0}|}{\ell }\right],$$
where the index *m* singles out the momentum eigenstates (I|m〉=m|m〉) and the system is initially prepared in the eigenstate $|m_{0}\rangle$. Therefore, for $t>t^{⋆}$ only
(9)$$\sqrt{\langle {\left(\Delta m\right)}^{2}\rangle }\approx \sqrt{Dt^{⋆}}\approx \ell$$
levels are populated.

An estimate of $t^{⋆}$ and *ℓ* can be obtained by means of a heuristic argument [28]. Since the number of levels involved grows diffusively, $\propto \sqrt{t}$, and, due to the Heisenberg principle, the discreteness of levels is resolved down to an energy spacing ∝1/t, then the discreteness of spectrum eventually dominates. The localized wavefunction has significant projection over about *ℓ* eigenstates of the Floquet operator *U*, which determines the evolution of the system over one map step. This operator is unitary and therefore its eigenvalues can be written as $\exp(i\lambda_{i})$, with $\lambda_{i}$ (known as quasienergies) distributed in the interval [0,2π]. The mean level spacing between quasienergy eigenstates which significantly determine the dynamics is ΔE≈2π/ℓ. The Heisenberg principle tells us that the minimum time required to the dynamics to resolve this energy spacing is given by
(10)$$t^{⋆}\approx 1/\Delta E\approx \ell .$$
Relations (Equation 9) and (Equation 10) imply
(11)$$t^{⋆}\approx \ell \approx D.$$
The quantum localization can be detected if *ℓ* is smaller than the system size *N*.

## 3. Quantum Computing of Dynamical Localization

We simulate dynamical localization with n=3 qubits on a real quantum hardware. The initial condition is peaked in momentum, $\psi_{0}\left(m\right)=\delta_{m,m_{0}}$, with $m_{0}=0$. The quantum algorithm for the sawtooth map allows us to compute the wave vector $\psi_{t}\left(m\right)$ as a function of the number of map steps, and then the momentum probability distribution $W_{t}\left(m\right)={\left|\psi_{t}\left(m\right)\right|}^{2}$. We consider k≈0.273<1, so that the distribution is already localized after a single map step. On the other hand, we use K=1.5, corresponding to diffusive behavior for the underlying classical dynamics. In Figure 2 we compare after t=1 map step the ideal, noiseless distribution (green curve) with the Qiskit simulator (blue curve), used to test the quantum algorithm before submitting it to a real quantum hardware (red curve). Data for the quantum hardware are for the recently released (8th January 2021) *lima* quantum processor. This machine has a nominal quantum volume $V_{Q}=8$ (i.e., a single number meant to encapsulate the quantum computer performance [29]), which is equal or smaller than the quantum volume of other freely available quantum processors. Nevertheless, in our simulation we obtained the best results for the localization problem with this machine (see discussion below).

As we can see in Figure 2, the quantum hardware exhibits a localization peak after a single map step. On the other hand, the height of the peak, $W_{1}\left(0\right)\approx 0.56$, is significantly smaller than the noiseless value, $W_{1}\left(0\right)\approx 0.83$, and the prediction of the Qiskit simulator, $W_{1}\left(0\right)\approx 0.74$. These results show that the Qiskit simulator may at least underestimate some of the relevant noise channels. It is also interesting to remark that, while the ideal distribution is symmetric around its peak, the noisy ones are markedly asymmetric. We interpret this asymmetry as a consequence of the binary coding of momentum eigenstates. For instance, a noise channel leading to a flipping of the most significant qubit would induce a probability transfer from the peak (m=0, corresponding to the state |100〉 of the three qubits) to the state |000〉 (corresponding to m=−4).

To compare the performance of different freely available quantum machines, we ran the sawtooth quantum algorithm for one map step for several days (from 24th February to 5th March 2021). In Figure 3, we show data for *lima*, *belem*, and *quito* (relase date 8th January 2021, nominal quantum volume $V_{Q}=16$), *santiago* (3rd June 2020, $V_{Q}=32$), *athens* (3rd March 2020, $V_{Q}=32$), and *ibmqx2-yorktown* (24th January 2017, $V_{Q}=8$). This latter is an old machine having a low quantum volume (nominally), but its bow-tie layout is in principle more favorable for the present application, as it allows direct connectivity between all the three qubits involved in the algorithm. Actually the unfavorable layout of the other machines is not very relevant for our 3-qubit simulations. Indeed, the number of extra quantum gates needed to relate non-physically connected qubits, in order to make the quantum algorithm work on these devices, is such that the execution time of the algorithm never exceeds the minimum decoherence time of the qubits used for all the machines. For such qubits, the algorithm run time for yorktown is ≈11.1 μs, while for the other devices requiring extra gates it increases only up to ≈16.7 μs on average. By comparing the height of the localization peak for several days of runs on the available quantum processors, we conclude that *lima* is to be considered the best performing machine at time of writing, both in terms of the best achieved absolute performance ($W_{1}\left(0\right)\approx 0.56$) and stability of the results obtained.

In order to quantitatively relate the effectiveness of the dynamical localization quantum algorithm to the error rate, we compare the height of the localization peak to the noise level in each of the analyzed devices, as measured from the day-to-day calibrations (nominally given on the IBM Q website). As a measure of the quality of the qubits, we define an average decoherence time $\langle T_{\mathrm{dec}}\rangle$, obtained after averaging over the n=3 qubits used in a given simulation, $T_{\mathrm{dec}}=\min\left\{T_{1},T_{2}\right\}$, where $T_{1}$ and $T_{2}$ are the standard relaxation and dephasing times, respectively. Then, we also estimate the overall relative error of a quantum computer run, defined as $E_{\mathrm{tot}}$, as the sum of the relative errors of each quantum gate and of the final readout error. Results are then reported in Figure 4, where we use a color code from red to violet to quantify $\langle T_{\mathrm{dec}}\rangle$, while $E_{\mathrm{tot}}$ is proportional to the size of the symbols. It can be seen that the height of the localization peak is larger for larger $\langle T_{\mathrm{dec}}\rangle$ and smaller $E_{\mathrm{tot}}$, i.e., corresponding to small violet circles (in particular, see data for the *lima* processor). On the other hand, the worst performances are obtained for smaller $\langle T_{\mathrm{dec}}\rangle$ and larger $E_{\mathrm{tot}}$, i.e., corresponding to large red circles (in particular, see data for the *yorktown* processor). However, large fluctuations around this average behavior can be noticed. This is expected, since the data from the Qiskit simulator that are obtained by assuming the same nominal parameters used to define $\langle T_{\mathrm{dec}}\rangle$ and $E_{\mathrm{tot}}$, largely underestimate noise, as it is shown in Figure 2. In particular, fluctuations of the qubit quality parameters between different calibration procedures, memory effects, and cross-talks between qubits (even not specifically employed in a given simulation) could also affect the performance of a quantum processor, and may not be fully captured by the currently implemented noise models.

Finally, with the aim of further testing the capabilities of quantum processors to simulate dynamical localization, we show in Figure 5 the localization peak value as a function of the number of kicks. For the noiseless case the distribution remains localized with fluctuations of $W_{t}\left(0\right)$ around 0.9. For the *ibmqx2 - yorktown* quantum processor the peak at m=0 does not emerge from noise already at t=2. On the other hand, for the *lima* quantum processor the peak is visible up to t=3 kicks. Evidently, the Qiskit simulator underestimates the level of noise in the corresponding devices, and predicts a peak decay in 3–5 map steps (for *ibmx2-yorktown*) and in 10–15 kicks for *lima*. Overall, the data of Figure 3 and Figure 5 testify the significant improvement of quantum hardware performance in the last few years. As a closing remark, it is worth reminding that error mitigation techniques can be applied to reduce the impact of various sources of noise on near term quantum processors [30,31,32]. We have applied an error mitigation procedure based on Ref. [31] in combination with the Qiskit noise model, and only found a limited effect in improving the specific quantum localization algorithm (results not shown). A more detailed characterization of noise in such devices is by itself an interesting problem, which goes, however, beyond the scope of the present work.

## 4. Conclusions

We have simulated the dynamical localization phenomenon of the quantum sawtooth map by means of several freely available IBM quantum processors, remotely accessed through cloud quantum programming. The localization peak emerges from noise, even though significantly reduced with respect to the noiseless dynamics, with up to three steps of the sawtooth map model for three qubits. Our results demonstrate a significant improvement of the performances of the available quantum processors over the past few years. This study shows that quantum computing of dynamical localization may be employed as a fast and convenient benchmark to check the performance of future generations of quantum processors. In particular, we could envision studying the maximum number *n* of qubits for which the peak height after one kick is reproduced with a given accuracy. Since the number of elementary quantum gates required to simulate a single step of the map scales as $n^{2}$, in this “localization test” both the number of qubits and the circuit depth grow with *n*. The results of this work show that, for n=3, the localization test does not necessarily agree with the quantum volume measure.

## Figures and Tables

**Figure 1 entropy-23-00654-f001:**
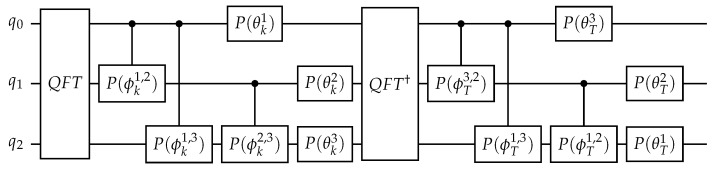
Quantum circuit of sawtooth map with n=3.

**Figure 2 entropy-23-00654-f002:**
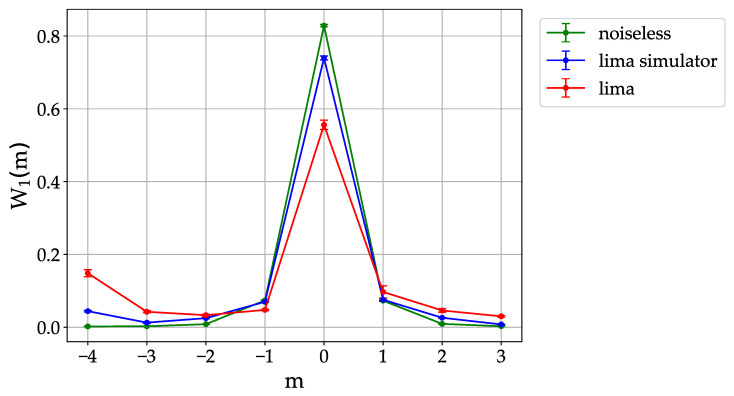
Dynamical localization in the quantum sawtooth map with n=3 qubits, K=1.5, k≈0.273 (T=2πL/N, with L=7). Hereafter data from the quantum processors (here *lima*) are obtained after averaging over 10 repetitions of 8192 experimental runs. Data taken on 26th February 2021.

**Figure 3 entropy-23-00654-f003:**
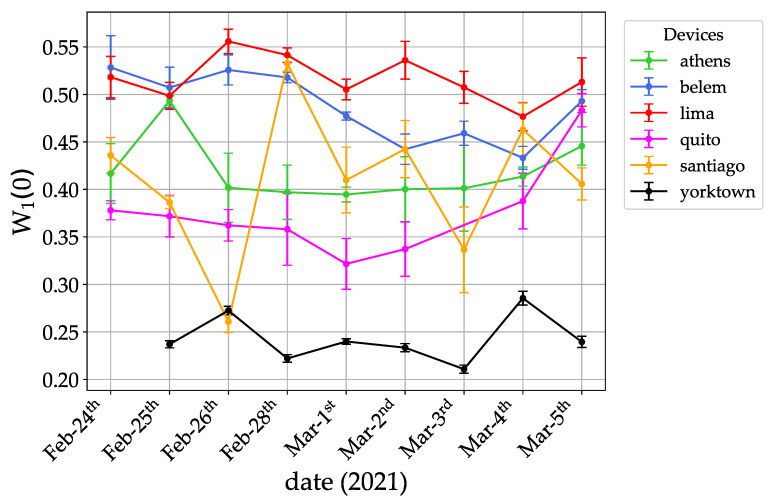
Height of the localization peak after one map step for several quantum processors. Data taken from 24th February to 5th March 2021.

**Figure 4 entropy-23-00654-f004:**
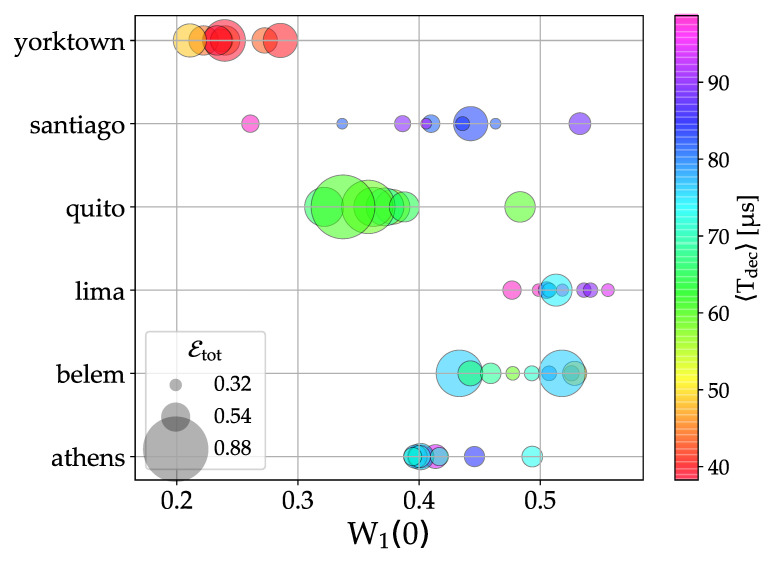
Height of the localization peak for the six quantum processors used in this work. For all the data from Figure 3, we show the average decoherence time, $\langle T_{\mathrm{dec}}\rangle$ (color coded from red to violet), and the overall relative error, $E_{\mathrm{tot}}$ (size of the circles), as defined in the text.

**Figure 5 entropy-23-00654-f005:**
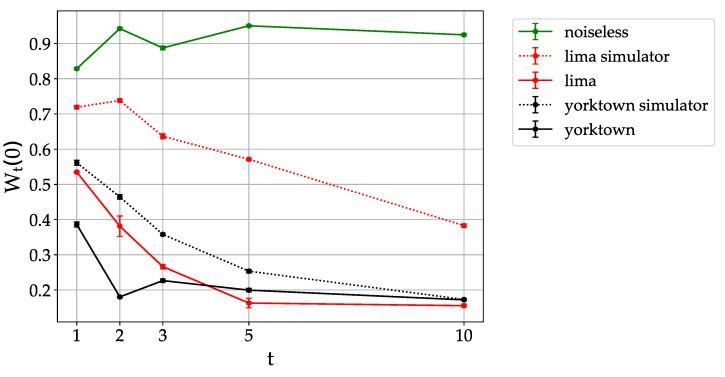
Height of the localization peak as a function of the number of map steps, for parameter values as in Figure 2. Data are for *lima* (red) and *ibmqx2-yorktown* (black). Data taken on 14th February (*ibmqx2-yorktown*) and 5th March 2021 (*lima*).

## Data Availability

The dataset used and analyzed in the current study are available from the corresponding author on reasonable request.

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
