# Peer review of "Dynamical Localization Simulated on Actual Quantum Hardware"

_entropy, 2021, doi:10.3390/e23060654_

Round 1

Reviewer 1 Report

The paper studies the quantum simulation of dynamical localization, and a quantum algorithm is provided for this simulation. This topic is of interest in a way, but I have the following points to be addressed before further consideration.

  1. The quantum algorithm is very confused to understand, and this should be improved: (i) Why is the quantum sawtooth map the most convenient model to simulate the dynamical localization? (ii) The correctness of the quantum algorithm should be proved strictly by verifying each step. (iii) The complexity of the algorithm should be analyzed carefully.
  2. Section 3 gives a simulation of dynamical localization with 3 qubits, the advantages over classical simulation should be pointed out.
  3. The main results should be presented or reformulated by means of a number of Theorems.

Author Response

Reviewer 1 acknowledges that the topic of our paper is "of interest". Then he/she raises three specific questions, which we address below. Preliminarily, we point out that it is not the purpose of the present paper to discuss the quantum algorithm for the quantum sawtooth map, since it was proposed and validated long ago in Refs. [26.27]. We instead focus on its implementation on actual quantum hardware. In view of that, while we discuss the main steps of this quantum algorithm for completeness, for further details the reader is directed to Refs. [26,27]. This point has been stressed at the end of the introduction to avoid possible misunderstanding.

Comment:
"The quantum algorithm is very confused to understand, and this should be improved: (i) Why is the quantum sawtooth map the most convenient model to simulate the dynamical localization? (ii) The correctness of the quantum algorithm should be proved strictly by verifying each step. (iii) The complexity of the algorithm should be analyzed carefully."

Our answer:
With regard to point (i) we have clarified that the decomposition used for U_k is possible only for the sawtooth map. For other models like the kicked-rotor model an efficient implementation of U_k was proposed (see the newly added Ref.[28]) taking advantage of auxiliary qubits. As a consequence, the overall cost of the simulation of such models is greatly enhanced with respect to the sawtooth map. 

With respect to point (ii), the decomposition of the evolution operator U into one- and two-qubit gates is exact, without any approximation involved. It is based on the one proposed long ago (Refs. [26,27]), with quantum gates here expressed in terms of those available in the IBM Qiskit interface. Moreover, we combined the gates where possible and avoided physical SWAP gates in favor of relabelling. That is, we optimized the algorithm of [26] for the given quantum hardware. 

As for (iii), the number of elementary gates required to implement the evolution operator for one map step is O(n^2), to be compared with O(2^n n) operations required to implement the same operation on a classical computer with the best known classical algorithm, based on the fast Fourier transform between the theta and the I representation. That is, dynamics of the quantum sawtooth map can be simulated on a quantum computer exponentially faster than any known classical algorithm.

The above points have been clarified in the text, see added sentences in Sec. 2.1 highlighted in blue. 

Comment
Section 3 gives a simulation of dynamical localization with 3 qubits, the advantages over classical simulation should be pointed out.

Our answer
The advantage of the quantum algorithm over the classical one as a function of the number of levels N=2^n is discussed in Sec. 2.1 (see also our answer to the point (iii) above). Sec. 3 instead fulfills the main purpose of our paper, that is, to check the ability of nowadays quantum computers to simulate dynamical localization. which is a complex quantum interference phenomenon. 

Comment
The main results should be presented or reformulated by means of a number of Theorems.

Our answer
We respectfully disagree with the referee on this point. The quantum algorithm for the sawtooth map was presented long ago in Refs. [26,27] and the reader may find further details therein. The implementation of such an algorithm on actual quantum processors, remotely accessed through cloud quantum programming, is described in Sec. 3. The discussion of these experimental data does not need to be reformulated by means of theorems. 

Reviewer 2 Report

The authors simulate the dynamic localization of the quantum sawtooth map on an actual quantum computer (actually several different QC's). They addressed this theoretically in Refs 26 and 27. One could object that this is mere demonstration, but I would differ. I think it's a real sign of progress that these ideas are being tested on actual hardware and I applaud this paper. I will admit that the differences shown in Figure 3 are sobering. But this is early days.

Figure caption for Fig. 1 is too hard to read with all the math in it. I would suggest elevating the math to the text.

Some space between the bullets in 66-80 would help readability too.

Author Response

We thank the reviewer for their comments. Indeed it is a real sign of progress that the quantum algorithm for the quantum sawtooth map, proposed twenty years ago in Refs. [26,27] can now be simulated on an actual quantum hardware, freely available for cloud quantum programming. Of course, reliable simulations for this specific model are currently limited to a small number of qubits (n=3), due to a significant impact of the different noise and decoherence sources. Nevertheless, we do believe it is important to monitor and benchmark progresses of quantum hardware by useful attempts, and our contributions makes a step in this direction.

The referee has two suggestions for changes, namely:

"Figure caption for Fig. 1 is too hard to read with all the math in it. I would suggest elevating the math to the text.

Some space between the bullets in 66-80 would help readability too."

We have followed these suggestions in the revised version of our paper. 

Reviewer 3 Report

In the manuscript “dynamical localization simulated on actual quantum hardware,” Pizzamiglio and collaborators propose using the problem of dynamical localization as a reference for certifying quantum hardware performances. For that, the Authors used the freely available IBM quantum platform to run and compare the results obtained for the quantum sawtooth map in several different devices (six in total). 

The paper is clearly written, and the presentation is scientifically sounding. I agree with the perspective brought by the Authors that the problem of dynamical localization may indeed furnish means to evaluate the performance of quantum devices since such a phenomenon is a manifestation of entanglement’s presence in the processor. 

The Authors claim that the height of the peak localization would constitute a good figure of merit for assessing the computer performance. Even though such a hint sounds plausible, no further investigation is presented in the manuscript besides comparing the results obtained from different machines (figures 3 and 4). That comparison could advocate in favor of the proposal if the level of noise estimated for each day/run would have been presented in the paper. With the information, one could infer if the reduction of the peak would represent a bona fide metric for stating that one computer had outperformed compared with others. In addition, there is no discussion on what kind of error the peak would represent algorithmically since the dynamical localization phenomenon seems detectable in several of the devices. Therefore, without establishing a clear relationship between the height of the peak and the computer performance, I feel that the presentation lacks its strength, and it would not surpass the status of a conjecture. Because of that, I would not recommend the paper for publication in its current form and suggest the Authors think and provide means for assessing the relation between performance and the height of the peak localization, showing that it can be considered a bona fide relationship. 

Author Response

We thank the Reviewer for their comments. Indeed, dynamical localization is a manifestation of the presence of entanglement in the processor. In the following, we report the Reviewer's question, asking to quantitatively relate the height of the localization peak to the level of noise in the devices analyzed. We have carefully followed the suggestion, which helped improving our manuscript, and below we give our detailed answer:

"The Authors claim that the height of the peak localization would constitute a good figure of merit for assessing the computer performance. Even though such a hint sounds plausible, no further investigation is presented in the manuscript besides comparing the results obtained from different machines (figures 3 and 4). That comparison could advocate in favor of the proposal if the level of noise estimated for each day/run would have been presented in the paper. With the information, one could infer if the reduction of the peak would represent a bona fide metric for stating that one computer had outperformed compared with others. In addition, there is no discussion on what kind of error the peak would represent algorithmically since the dynamical localization phenomenon seems detectable in several of the devices. Therefore, without establishing a clear relationship between the height of the peak and the computer performance, I feel that the presentation lacks its strength, and it would not surpass the status of a conjecture. Because of that, I would not recommend the paper for publication in its current form and suggest the Authors think and provide means for assessing the relation between performance and the height of the peak localization, showing that it can be considered a bona fide relationship."

This is indeed a very interesting question, for which we have added and thoroughly discussed 
a new figure (Fig. 4) to address this crucial point. The comparison can be at the level 
of the relaxation time of the qubits, T_1, and their dephasing time, T_2; in each device, we specifically considered only the three qubits used for the simulation of dynamical localization. We are then able to show that the height of the localization peak increases with the minimum decoherence time (i.e., we define minimum between T_1 and T_2), averaged over the three qubits, which we introduce as a new figure named 'average decoherence time'. 
Moreover, we show that the peak height decreases with the total relative error of the simulation, which we have roughly estimated as the sum of the relative errors of the performed quantum gates  and of the final measurement error (as nominally given for each of the quantum devices on the IBM website). As the Referee might appreciate, the general trend of Fig. 4 shows that the quality of the quantum simulation decreases with the noise level, thus quantitatively confirming our original claims. On the other hand, there are large fluctuations around the average behavior. This is largely expected because, as shown in Figs. 2 and 5, the noise model provided by IBM and integrated in their online simulators of real quantum devices clearly underestimates the noise. That is, there are other noise channels affecting the simulation whose effects are not included in the model, quite evidently. Such effects might be conjectured as arising from cross-talks, memory effects, fluctuations of the device parameters between calibrations that are not properly taken into account. While a more detailed characterization of noise in such devices is by itself an interesting problem, this lies, however, beyond the scopes of the present paper. 
We have also added a final discussion to highlight that error mitigation techniques can be applied and have been developed to (partly) heal the effects of noise on quantum computations performed on such near-term quantum devices. We have tried to apply such error mitigation techniques (at least, the ones integrated within the Qiskit package), and we have noticed that they only have a limited effect in improving this specific quantum algorithm. For this reason, and considering that the present work aims at evidencing the role of noise on the reduction of entanglement on increasing quantum register size, we have decided not to further discuss these results, which are not reported in the manuscript.

Round 2

Reviewer 3 Report

I think the new version provided by the Autorhs has improved the presentation and substantiated the claim that the height of the peak localization could be considered a good figure of merit for determining computer performance. Indeed, the new figure (figure 4 in the new version) provides pertinent elements for characterizing the possible relationship between the peak's height and the computational success. Even though the data presented does not show an indisputable connection between those, I think it provides good evidence that it is worth investigating and considering such a relationship. As the Authors pointed out, their current access to the machines is limited and does not allow them to investigate further the fluctuations observed. Nevertheless, I think the paper deserves publication now and can foment new investigation on this subject.